## REVIEW ARTICLE

# When dormancy fuels tumour relapse

Karla Santos-de-Frutos[1] & Nabil Djouder [1✉]

Tumour recurrence is a serious impediment to cancer treatment, but the mechanisms involved are poorly understood. The most frequently used anti-tumour therapies—chemotherapy and radiotherapy—target highly proliferative cancer cells. However non- or slow-proliferative dormant cancer cells can persist after treatment, eventually causing tumour relapse. Whereas the reversible growth arrest mechanism allows quiescent cells to re-enter the cell cycle, senescent cells are largely thought to be irreversibly arrested, and may instead contribute to tumour growth and relapse through paracrine signalling mechanisms. Thus, due to the differences in their growth arrest mechanism, metabolic features, plasticity and adaptation to their respective tumour microenvironment, dormant-senescent and -quiescent cancer cells could have different but complementary roles in fuelling tumour growth. In this review article, we discuss the implication of dormant cancer cells in tumour relapse and the need to understand how quiescent and senescent cells, respectively, may play a part in this process.

Despite our growing knowledge of tumour biology and genetics, cancer remains a deadly disease. A high percentage of treated patients relapse after surgery or adjuvant therapies, and the tumour cells involved in the relapse often exhibit increased tumour propagating potential, manifested as local or distant disease recurrence. However, the mechanisms of tumour recurrence are largely unknown.

In addition to their genetic modifications, tumours comprise heterogeneous masses of cells that may differ in their capacity to support tumour growth, metastasis or resistance to therapy[1]. A growing tumour mass may consist of millions of proliferating cells, but also of some non- or slow-proliferative cells that are not sensitive to anti-proliferative therapies. Resistant dormant cells could fuel tumour regrowth after disease remission. However, our knowledge of the biology of dormant tumour cells is cripplingly limited. The recent identification of therapy-resistant cell populations with dormancy potential in both solid and hematologic tumours, including melanoma[2], glioblastoma[3], leukaemia[4] and pancreatic[5,6] and ovarian[7] cancers suggests that these dormant populations, resistant to cancer treatments, play a role in tumour relapse. Furthermore, dormant quiescent cancer cells, also referred as slow-proliferating or slow-cycling cancer cells throughout this review—which stall in $G_0$ phase or rarely enter the cell cycle, and/or senescent cancer cells in tumours could contribute to therapy resistance and tumour recurrence (Fig. 1)[2,8]. However, solid in vivo evidence of persistent tumour cells involved in tumour relapse are lacking and the molecular mechanisms behind such recurrence are largely unknown. Development of new genetic mouse models to track dormant cells would help to better understand how dormancy could fuel tumour relapse.

The tumour-initiating ability of dormant cells, their capacity to self-renew and ability to differentiate into various tumour bulk subpopulations led the scientific community to think about the involvement of cancer stem cells (CSCs) in tumour relapse[9]. The general feature of CSCs is their ability to initiate tumour outgrowth, and several similarities are shared between the theory of CSCs of tumour development and the concept of cancer dormancy[10]. CSCs, like dormant cancer cells, survive conventional cancer therapies and can evade anti-tumour immune

[1] Molecular Oncology Programme, Growth Factors, Nutrients and Cancer Group, Centro Nacional de Investigaciones Oncológicas (CNIO), Madrid, Spain. ✉email: ndjouder@cnio.es

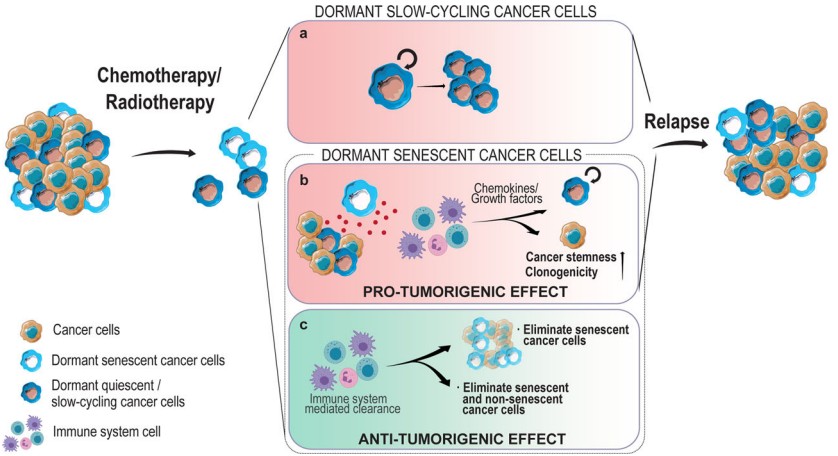

**Fig. 1 Schematic representation of the hypothesis of the intratumoural heterogeneity and the effects of anti-proliferative therapies.** Most frequently used anti-proliferative therapies are meant to eliminate rapid-proliferative cancer cells. The remaining dormant cell-driven relapse mechanisms differ depending on whether the cells involved are quiescent/slow-cycling or senescent. **a** Dormant-quiescent/slow-cycling cells can re-enter the cell cycle in response to appropriate microenvironment changes or SASP signals secreted by senescent cells. Dormant senescent cancer cells can have pro- or anti-tumorigenic effects mainly depending on the SASP content and hence, recruiting immune cells. **b** Although dormant-senescent cells have apparently undergone irreversible growth arrest, their SASP secretion induces slow-cycling cell proliferation, mainly mediated by immune cell recruitment, and induces clonogenicity and cancer stemness in neighbouring cells. **c** Alternatively, immune system cells recruited by SASP may eliminate senescent cancer cells, or eliminate both senescent and non-senescent cancer cells, causing tumour eradication[39].

responses. However, several lines of evidence suggest that CSCs can consist of distinct heterogeneous subpopulations, including fast-cycling or slow-cycling/quiescent subpopulations[11]. The quiescent subpopulation could be directly linked to dormant cancer cells, and might therefore exploit latency state to ensure long-term tumour maintenance upon critical environments. Thus, CSCs could be considered as quiescent subpopulations critical in the switch from dormancy to proliferation state to promote tumour outgrowth. Based on their similarities, eradication of dormant cells could also be translated into strategies to eliminate slow-cycling CSCs to eventually minimize the risk of cancer relapse. Further insights into the CSC biology could help us to better understand the mechanisms underlying cancer cell dormancy.

Here, we discuss and stress the need to elucidate the roles of dormant quiescent/slow-cycling and senescent cancer cells, which represent a therapy-resistant cell population reservoir in tumour relapse, often occurring after few months and even several years in the absence of appreciable tumour following therapies. Moreover, mechanisms by which this dormant cell population persist and survive cancer treatment will be discussed.

## How do dormant cells resist anti-cancer therapies?

Surgery, chemotherapy, and radiotherapy are the most commonly used cancer treatments, despite immune, hormonal, and targeted therapies are becoming more frequently used. Angiogenesis inhibitors, proteasome inhibitors, small-molecule inhibitors (such as tyrosine kinase, mTOR or PARP inhibitors), monoclonal antibodies (such as EGFR or HER2 inhibitors), and drugs that target histone deacetylases or retinoic acid receptors are among the currently used targeted therapies. Cancer chemotherapeutic agents are commonly categorized into cytotoxic and cytostatic drugs, which typically kill both healthy and cancer cells, and genotoxic agents, which directly or indirectly induce DNA lesions and damage[12]. Radiotherapy uses ionizing radiation, which also directly affects DNA structure by inducing DNA strand breaks, particularly, double-strand breaks[13]. Chemo- and radiotherapy both target rapidly dividing cells, causing their death.

Despite the existence of many different cancer chemotherapeutic drugs and, targeted and efficient anti-cancer

radiotherapy prescribed after surgery, a high percentage of patients experience relapse after months or even years of treatment discontinuation. Depending on the type of cancers, tumour recurrence is generally considered to occur early or late. In the case of acute lymphoblastic leukaemia most studies classify early tumour relapse as occurring between the first 18–36 months from diagnosis, and late relapse those as occurring after 36 months[14,15]. On the other hand, breast cancer recurrence is categorized as early when tumour recurrence occurs before 5 years of diagnosis and late after 5 years' time[16,17]. In general terms, early recurrence is more prone to occur, while late relapse is thought to be developed due to a long-term dormancy. The probability that a tumour relapse clearly depends on the cancer type. For instance, 20–40% of breast cancer patients[18] and 50–70% of hepatocellular carcinoma (HCC) patients[19] develop recurrence over a 5-year period, and relapse is almost inevitable in glioblastoma patients[20]. Why does this occur? Since most chemotherapeutic agents and radiotherapy treatments are designed to eliminate rapidly proliferating cancer cells, tumour relapse may be driven by non- or slow-cycling dormant resistant cells within the tumour that give rise to tumour dormancy[21] (Fig. 1). For example, in the case of radiotherapy, glioma cells resistant to ionizing radiations reportedly display increased DNA damage-response mechanisms, inducing therapy refractivity[22]. Moreover, recent in vitro studies mimicking aromatase inhibitor-induced resistance have identified a so-called pre-adapted cell population which triggers a dormant or sleeper state resistant to therapy, facilitating tumour relapse[23].

Different consensual models have been proposed to explain the survival of residual or dormant cancer cells, most of which are based on pre-exiting rare sub-clones that carry mutations conferring resistance to therapies. However, numerous recent studies are elevating the importance of non-mutational mechanisms and propose that mutation-induced resistance could not be the main mechanism leading to dormancy[24,25]. Interestingly, observations suggest that dormancy can be an adaptive strategy for cancers during times of stress[26] and in cases where undetectable residual cancer cells make the patient asymptomatic. Recent studies based on cellular barcoding on colorectal and breast cancer cells suggest

that dormant cells are not a pre-exiting population in tumours, but that cancer cells have an equipotent ability to enter the dormant state, similar to the embryonic diapause[24,25]. The origin of dormant cancer cells in tumour relapse remain elusive; whether these dormant cells pre-exist in the tumour and chemotherapy promotes their selection, or cancer therapies induce their transition to a dormant state in a subpopulation of cancer cells still needs to be confirmed.

## Quiescent cells vs. senescent cells

Cellular dormancy is often defined as a non-proliferating state of a cell, but commonly discussed in terms of two growth arrest mechanisms: quiescence, in which cells are in a non-proliferative or slow-cycling state, with a reversible growth arrest, and senescence, in which cell cycle arrest is largely irreversible[10,27,28]. The mechanisms of tumour relapse induced by reactivation of dormant cancer cells depend on whether the cells became dormant via quiescence or senescence.

**Dormant quiescent slow-cycling cancer cells**. As noted above, quiescence is considered a reversible state in which a cell ceases to divide but retains the ability to re-enter the cell cycle. It is generally believed that quiescence is the most appropriate mechanism for describing cellular dormancy[10,27]. In particular, dormancy has been demonstrated to represent a special case of quiescence among stem cells[29]. Quiescence is a cellular process that preserves stem cell function in case it is needed in tissue homeostasis or repair, and shares feature with senescent cells[30,31]. Such dormant quiescent cells are also known as slow-cycling or slow-proliferating cells because they stall in $G_0$-$G_1$ phase or rarely enter the cell cycle. Quiescent cells are arrested in the $G_0$-$G_1$ phase, meaning that the activity of cyclin-dependent kinases (CDK) is reduced, while the activity of the CDK inhibitor p27, which regulates the transition from $G_0$ through $G_1$ into S phase, is elevated[32,33]. In response to injury, quiescent stem cells transit between the $G_0$ phase and an 'alert' phase called $G_{(Alert)}$, a process controlled by mTORC1. $G_{(Alert)}$ represents an adaptive mechanism to respond rapidly to damaging reagents and stress, priming stem cells for a rapid cell cycle entry to repair the injured organ[34]. Interestingly, the $G_0$ phase is characterised by low metabolic activity, with a decrease in the production of ribosomal RNA and proteins, leading to reduction of their volume and size[35]. Recent studies have suggested that quiescent cells could have an embryonic diapause-like state in breast and colorectal cancers. This diapause-like state is defined by decreased mTOR activity, leading to increased autophagy, suggesting that chemotherapy combined with autophagy inhibitors could be efficient to kill these quiescent cancer cells[24,25]. However, a deeper analysis of their transcriptomic signature demonstrates that this quiescent cancer cells might be distinct from the diapause state described for embryos, but rather resemble the paused embryonic stem cells[24]. Clearly, quiescent cells might adopt different states of dormancy which should be further characterized by developing genetic tools allowing their labelling and track in vivo during tumour recurrence, and by single-cell RNA-sequencing methodology.

Whereas highly proliferative cells promote DNA replication stress-driven mutations, the quiescent state seems to enable cancer cells to acquire new somatic mutations essential for disease progression. In fact, quiescent cells express the lower levels of genes involves in DNA damage repair mechanisms[36]. Moreover, the preferential use of the more error prone non-homologous end joining-mediated DNA repair mechanism rather than homologous recombination renders quiescent cells more susceptible to suffer genomic instability and transformation upon DNA

damage[37]. These new mutations might facilitate quiescent cancer cells to escape the immune system. Agudo et al. demonstrated that slow-cycling cells are immune-protective, a mechanism that is not specifically shared by rapid proliferating cells[11]. Furthermore, several studies on quiescent cells have suggested that immune evasion could be obtained through neo-antigen loss. Recent single-cell RNA-sequencing analysis on HCC suggested that the loss of main clonal neo-antigens during relapse could explain the impossibility of CD8+ T cells to recognize cancer cells and induce their clearance[38]. Moreover, new mutations in quiescent cells may lead to new sub-clonal neo-antigens, escaping the memory T cells.

Natural killer (NK) cells are known to be implicated in tumour cell clearance by the secretion of several inflammatory cytokines, eliminating quiescent and senescent cancer cells. Iannello et al. demonstrated that NK cells are recruited by the secretory phenotype of senescent cells mediated by p53[39]. An example of immune evasion involves the cell surface glycoprotein UL16 binding protein 1 (ULBP1), a member of the MHC class I superfamily, which is expressed on the surface of malignant transformed cells[40]. ULBP1 functions as a stress-induced ligand for NKG2D receptor, activating NK cell-mediated cytotoxicity[40–42]. Recent studies in breast cancer and lung adenocarcinoma suggest that slow-cycling or quiescent cells downregulate ULBP ligands (ULBP1-5), thereby inactivating NK cells and allowing the CSCs to repopulate the tumour niche[43]. The authors demonstrated that expression of DKK1, an autocrine WNT inhibitor, results in the downregulation of NK activating ligands and death signal receptors. DKK1 depletion leads to NK cell-mediated cytotoxicity of quiescent cells in vitro, but the authors did not show that ULBP ligand is critical for NK evasion of quiescent cells. The reversibility of growth arrest in quiescent cells may thus enable slow-cycling cancer cell-mediated tumour relapse in cases where microenvironmental changes enable these dormant cells to resume normal cell cycle behaviour (Fig. 1).

Several studies have focused on deciphering the involvement of quiescent cells in tumour relapse. It was proposed that recurrence of basal cell carcinoma after vismodegib treatment was due to a switch of proliferative Lgr5-expressing cells to quiescent cells that become non-targeted by vismodegib, leading to tumour relapse[21]. Another study reached the same conclusions and proposed a similar phenotypic switch in basal cell carcinoma leading to tumour relapse. The authors demonstrated that quiescent cells were able to re-enter a proliferative state after vismodegib discontinuation promoting tumour regrowth[44]. Furthermore, a rare quiescent cancer stem cell pool was identified in squamous cell carcinoma that becomes enriched following 5-FU treatment and displays increased tumour propagating potential. The quiescent stem cell pool co-existed with proliferative cells and transcriptomic analysis suggested that the dynamic transition between quiescent and proliferative states was mainly controlled by pro- and anti-proliferative cancer signalling factors, such as TGF-β. Further, TGF-β was the crucial factor directing quiescence in squamous cell carcinoma[45]. More studies are clearly needed to understand how slow-cycling cells escape from the immune system and re-enter in the cell cycle to promote tumour relapse. Moreover, despite the studies suggesting that tumour relapse is due to slow-cycling cells which persist after cancer treatment, further work is needed to fully demonstrate that tumour recurrence indeed relies on non-targeted quiescent cancer cells. Animal models to track these cells during tumour recurrence are urgently required to demonstrate the role of slow-cycling cells in tumour relapse. Moreover, the presence of senescent cells resistant to the mentioned therapeutic agents and their possible implication in recurrence[21,44,45] cannot be excluded and will be discussed below.

**Dormant senescent cancer cells**. By contrast, senescent cells are irreversibly arrested in the $G_1–G_1/S$ phase[46,47]. The cellular senescence programme can be activated by a wide range of extrinsic and intrinsic stressors, which eventually lead to activation or expression of the tumour suppressors p53 and/or p16INK4A[48–51]. Serrano et al. showed that Ras-mediated senescence requires p53 and p16INK4A/Rb to promote cell cycle arrest[49]. Telomere damage, oxidative stress and DNA damage, among others, activate p53, which induces p21 expression to inhibit the cyclin E-Cdk2 and promote cell-cycle arrest[52,53]. p16INK4A, in contrast, which can be activated by various oncogenes, epigenetic stress or nucleolar stress, inhibits cell-cycle progression via the disruption of the cyclin D-Cdk4/6 complexes[48,52,54]. p53 or p16INK4A-related pathways impede RB phosphorylation and hence, its inactivation. In turn, this leads to inhibitory binding to E2Fs transcription factors, thus preventing the expression of genes involved in cell proliferation and DNA replication[49,55]. Senescent cells are also metabolically very active, displaying an increased biomass. This high activity is needed to secrete stress-mediated granules[51,56]. Accordingly, senescence-mediated lysosomal compartment expansion leads to an increase SA-β-galactosidase or β-D-galactosidase activity[57], commonly used as a senescence biomarker[58]. Moreover, senescent cells exhibit an altered chromatin structure called senescence-associated heterochromatin foci (SAHF) which stains densely with DAPI and is enriched for histone modifications, mainly lysine 9-trimethylated histone H3. SAHFs play a role in the senescence-associated cell growth by sequestering and silencing proliferation-promoting genes, including the E2F target gene cyclin A[55].

Cellular senescence is thus a state of permanent cellular growth arrest induced by damage or stress. The detection of senescence markers in dormant cancer cells suggested that senescence may be another mechanism driving cellular dormancy. This idea is supported by two studies[59,60], reporting that BMP7 and SPARC, respectively, maintain prostate cancer cells in dormancy by inducing senescence[59,60]. The authors showed that when culturing metastatic prostate cancer cell lines in the presence of conditioned media from human bone marrow stromal cells, senescence-associated markers were upregulated. Particularly, they demonstrated in vitro that bone stromal cell-secreted BMP7 induces senescence by activating p38 MAPK signalling, in turn increasing the level of p21, which mediates the upregulation of the metastasis suppressor gene NDRG1 expression, ultimately resulting in cell-cycle arrest or dormancy[59]. p38 is known to be involved in cell cycle arrest regulation and plays a crucial role in the induction of senescence in response to a variety of stresses[61].

Despite senescence being considered a state of irreversible growth arrest, it is estimated that 1 in $10^6$ senescent cells could escape from senescence and re-enter the cell cycle[62]. Studies in non-small cell lung cancer cell lines suggest that chemotherapy-induced senescent arrest can be reversible in a small subset of cells, which mainly escape through the upregulation of Cdk2/Cdk1. The authors showed that 3–4 weeks after removal of the chemotherapeutic drug camptothecin, some cells were able to form colonies[62]. Furthermore, SPARC, a matrix-associated protein expressed and secreted by prostate cancer cells, induces dormancy of bone cells, a process sustained by SPARC-mediated activation of BMP7 secretion. Depletion of SPARC reawakens these dormant cells, leading to their growth[60]. Interestingly, Milanovic et al. demonstrated that a rare fraction of senescent cells could spontaneously be released from senescence and re-enter the cell cycle, giving rise to the so-called "post-senescence" state. The authors suggest that these "post-senescent" cells retain stem cell-related features (also known as senescence-associated stemness), suggesting a more aggressive behaviour and favouring

tumour relapse[30]. Yet, senescence reversibility seems an infrequent event. In support of this idea, Takahashi et al. suggested that blockage in cytokinesis could be a second barrier for cellular senescence, where p16INK4a-Rb pathway and senescence-associated chromatin remodelling support the irreversible cellular arrest, limiting senescence plasticity and implying the infrequency of this event[63].

The general properties of senescent cells may suggest that their main role in tumour recurrence does not involve reinstating the cell cycle. Instead, it is more likely to be driven by the release of secretory factors from senescent cells, which may modulate the microenvironment and particularly the behaviour of nearby immune cells. Immune system modulation is mainly driven by cytokines, chemokines, matrix remodelling proteases, and growth factors secreted by senescent cells exhibiting the so-called senescence-associated secretory phenotype (SASP)[8,64–66]. DNA damage leads to SASP programme activation, which is carried out by stress-response kinases. SASP or growth factors secreted by senescent cells could activate slow-cycling cells' proliferation in a paracrine way and/or via immune system activation, leading to tumour relapse (Fig. 1). In addition to the presence of cancer cells and stroma, innate (such as macrophages, neutrophils and NK cells) and adaptive immune cells (T and B lymphocytes) form part of the tumour microenvironment[67]. All these cell types communicate via autocrine and paracrine signals mediated by several immune modulators, such as chemokines and cytokines. The cellular diversity within the same inflammatory niche, the activation states of these various cell types, as well as the class and expression levels of the immune modulators will determine the pro- or anti-tumorigenic effects of the SASP[68]. Depending on this response, SASP can lead to the clearance or the protection of cancer cells, favouring cell dormancy and tumour recurrence. For instance, various physiological processes, such as increased cell survival, angiogenesis and suppression of anti-tumour adaptive immune responses are regulated by leucocyte infiltrates[67]. Moreover, the transcription factor NF-κB, a key mediator of inflammatory responses, regulates the expression of genes involved in the suppression of tumour cancer cell death, activates tumour cell cycle progression and stimulates epithelial-to-mesenchymal transition and angiogenesis[69]. SASP factors could therefore activate the pro-tumorigenic inflammatory response and thus, via the activation of inflammatory cells, promote surrounding cells' (slow-cycling cells) to proliferate. This cross-talk between senescent cells and slow-cycling cells enables the latter to become highly proliferative following paracrine signal activation. Consequently, senescent cells retain tumour propagation potential and can drive tumour re-initiation after chemo or radiotherapy. Pre-malignant senescent hepatocytes were found to accelerate the growth of HCC cancer cells in mice and humans mainly through SASP secretion-mediated immune recruitment[70,71]. Further, senescent cell-secreted IL-6 promotes reprogramming of the surrounding cells in vivo[72] and conditioning with senescent cell media promotes clonogenicity and cancer stemness in multiple myeloma cell lines[73], and to enrich chemotherapy-resistant cell populations in vitro in malignant pleural mesothelioma cell lines[74].

The presence of senescent cells may also favour other physiological processes, such as wound healing[31], embryonic development[75,76] and maturation of β cells[77]. On the other hand, cellular senescence contributes to non-cancerous pathologies. The accumulation of aberrant senescent cells generates an inflammatory niche, which might induce tissue damage and the development of various diseases, such as liver and lung fibrosis, diabetes, atherosclerosis and osteoarthritis[46,78,79]. Interestingly, the elimination of senescent cells improves these pathologies and contributes to longevity[46,78,80–82]. Liver fibrosis can be an

example of the role played by senescence in disease progression, which clearly depends on the cell type undergoing senescence and the inflammatory milieu generated. The general idea is that senescent hepatocytes and cholangiocytes are associated with fibrosis progression, most likely through paracrine signals activating hepatic stellate cells (HSCs) which are implicated in the production of the extracellular matrix of fibrotic scars[83]. However, senescence of HSCs can induce fibrosis regression by enhancing the expression of the matrix metalloproteases with fibrolytic activity, enabling the tissue to recover, and hence limiting liver fibrosis[84]. Moreover, senescent HSCs can modulate an immuno- surveillance response to promote their clearance via the activation of the NK cells, leading to the resolution of fibrosis[84].

The abovementioned findings clearly support the idea that quiescence and senescence are associated with different forms of dormancy that lead to distinct phenotypes capable of driving tumour relapse (Table 1). This complexity reinforces the necessity to better elucidate the mechanisms by which slow-cycling and senescent cancer cell populations participate in tumour relapse.

### The duality of senescent cells: anti- or pro-tumorigenic?

Senescence is considered a stress response induced by several intrinsic and/or extrinsic factors and mechanisms, such as che-motherapeutic agents, hypoxia, oncogene activation, aging and dysregulation of growth factors. Various studies have suggested that classical cytotoxic therapies, molecularly targeted therapies and immunotherapies can all trigger so-called "therapy-induced senescence" (TIS), converting tumour cells into senescent cells[64]. Remarkably, the widely used chemotherapeutic agent doxor-ubicin, which affects DNA structure, can enlarge the senescent cancer cell pool[85]. ATRX as a key regulator of TIS and indeed, both DNA-damaging agents, such as chemotherapeutic drugs and CDK4 inhibitors require ATRX expression and subsequent sup-pression of the *HRAS* locus to promote senescence induction. ATRX-depleted cell lines enter quiescence, following treatment with chemotherapeutic agents and CDK4 inhibitors[86].

Whereas the literature has mainly focused on chemotherapy-induced senescence, radiotherapy can also induce senescence in cancer cells[87–89]. TIS can have a profound impact, particularly in fractionated radiotherapy regimens where the radiation dose is increased incrementally. Because each dose of ionizing radiation will convert some tumour cells into senescent cells, the treatment may not have the expected anti-tumour effect by the time the

patient receives the highest doses. Unlike in apoptosis, cells that enter senescence are not killed; they remain in the tumour and retain metabolic and secretory activity despite not undergoing cell division[90]. Moderate doses of camptothecin convert 85–90% of non-small cell lung adenocarcinoma cell lines to senescent cells, while etoposide induces 40–60% and cisplatin 10–30% of cells to enter senescence. These senescent cells are identified by flattened morphology, increased cytoplasmic granularity, SA-β-galactosidase expression and reduced proliferation[62]. Further-more, other studies of chemotherapy-treated breast cancer patients tumour samples revealed that 41% of treated samples were positive to SA-β-gal[91]. Hence, although, TIS can also be detected in treated patients, senescent cells may have either pro- or anti-tumorigenic effects depending on their cellular or pathophysiological context and their production of secretory factors or SASP, which, as discussed further below, have pleo-tropic functions and is a two-edged sword in cancer[8,28,64,65].

### Anti-tumorigenic effects of senescent cells

Not only can senescence-associated cell cycle arrest inhibit tumour growth and progression[8,64], but the associated SASP can also modulate and reshape the tumour microenvironment to stimulate immune-mediated clearance of senescent cancer cells. SASP factors have different biological activities, and dynamic SASP patterns have been observed. The senescence process appears to have at least two distinct secretory phases in which different subsets of factors are secreted with opposite effects. The "first wave" or phase is mediated by cell-to-cell contact (juxta-crine) between senescent and neighbouring cells via the activation of NOTCH, and which leads to cell-intrinsic and extrinsic effects. NOTCH signalling pathway relies on ligand-dependent activation (JAG1/2 and DLL1/3/4 in humans) and it has to undergo a series of proteolytic cleavage steps, leading to the formation of NOTCH intracellular domain (NICD). NICD can translocate to the nucleus, where it induces the transcription of NOTCH target genes, thereby promoting "lateral senescence" or "paracrine senescence" of neighbouring cells[92]. In this regard, NOTCH modulates the expression of inflammatory cytokines, including the critical SASP factor TGF-β, which reinforces the paracrine senescence through p21-mediated cell cycle arrest[93]. Likewise, the transmission of senescence to neighbouring cells via a paracrine signal sets-up a tumour-suppressive function[94,95]. The "second wave" secretome is usually rich in C/EBP-β-dependent SASP with pro-inflammatory, fibrolytic and immune clearance properties. C/

---

**Table 1 Main differences between dormant senescent and quiescent cancer cells and their roles in tumour relapse.**

|  | Quiescent cancer cell | Senescent cancer cell | References |
|---|---|---|---|
| Cell cycle arrest | Reversible: $G_0$-$G_1$ phase arrest | Irreversible: $G_1$-$G_1$ / S phase arrest / Cytokinetic block | 32,33,46,47,63,139 |
| Markers | None | • $p16^{INK4}$ expression / p53 activity<br>• SASP factors<br>• SA-β-gal staining<br>• DNA damage-response<br>• γH2AX foci and SAHF formation | 46,49,58 |
| Effectors | p27 | p53 (and p21) and / or $p16^{INK4}$ mediated RB activation | 49–51 |
| Metabolic activity | Low (reduction in volume and size) | Very active (increased biomass leading to SASP production) | 33,51,56 |
| Role of immune system | Immune evasion | Attract immune cells by SASP secretion | 8,43,65,70 |
| Mechanisms of relapse | Re-enter cell cycle | Microenvironment modulation and immune cell recruitment via SASP | 21,30,44,68,70,100,101 |
| Structural changes | Chromatin compaction by methylation in H4K20 | SAHF formation and γH2AX foci / Lysosomal compartment expansion | 55,57,140 |

*RB* retinoblastoma protein, *SA-β-gal* senescence-associated β-galactosidase, *SAHF* senescence-associated heterochromatic foci, *SASP* senescence-associated secretory phenotype.

EBP-β induces the expression of inflammatory cytokines, such as IL-6 and IL-8, which attract and activate a wide range of immune system cells (e.g. CD8+ cytotoxic T-cells, B-cells, neutrophils)[96], favouring immune-mediated elimination of senescent cells[65]. The existence of a coordinated response of innate immune components required for the clearance of senescent cancer cells has been suggested[94,95,97]. A study of hepatocarcinoma provided an example of this type of anti-tumorigenic senescence, demonstrating that p53 loss is required to maintain the aggressiveness of cancer cells and its restoration induces senescence, immune recruitment and tumour cell clearance[94]. In addition, immune-mediated clearance of pre-malignant senescent hepatocytes is mainly driven by CD4+ T-cell based adaptive immunity by the secretion of diverse chemo- and cytokines, which also requires the activation of monocytes and macrophages[95]. Moreover, senescence-associated cell cycle arrest can inhibit tumour growth and progression[8,64]. Studies in *KRAS* mutant models of lung cancer demonstrated that the combination of MEK and CDK4/6 inhibitors lead to TIS, whereas components of SASP attracted NK cells, contributing to tumour regression[98]. The authors extrapolated their findings to poorly vascularized pancreatic ductal adenocarcinoma, and demonstrated that combinatory targeted therapies triggered senescence and in turn, SASP remodelled the tumour microenvironment and vascularity, increasing blood vessel density and permeability in order to facilitate chemotherapeutic agent uptake within the tumours and increased T cell infiltration, rendering it susceptible to immune checkpoint inhibitors[99]. Despite these findings suggesting that senescence can support tumour-suppressive mechanisms to restrict the development of malignant cells, they do not exclude that after a certain time, these senescent cells could negatively remodel the tumour microenvironment to favour tumour relapse once treatment is ceased.

**Pro-tumorigenic effects of senescent cells**. In some cases, SASP can also inflame the tumour microenvironment and accelerate tumour progression[8,28,64,65], probably depending on the SASP as well as on the type of immune cells composing the inflammatory milieu, which could influence proliferation and growth of cancer cells or activate the invasive properties of cancer cells including migration and angiogenesis[30,100]. SASP could also impair the immunosurveillance response by inhibiting the immune-mediated clearance, thus enabling cancer recurrence[70]. This ability of senescent cells to modify the microenvironment and the surrounding cells in a non-autonomous manner adds further complexity to tumours. Studies on several types of cancer have suggested that cellular senescence and SASP are barriers to complete tumour eradication, even though senescence has often been regarded as an intrinsic tumour suppressor mechanism like apoptosis[8,64–66]. Krtolica et al. were among the first to suggest that senescence may exhibit evolutionary antagonistic pleiotropy, which means that can have both beneficial and deleterious effects. They showed that soluble and insoluble factors secreted by senescent fibroblasts caused pre-malignant and malignant epithelial cells to proliferate and form tumours[100]. Furthermore, in a liver cancer mouse model, myeloid cells recruited by SASP factors released from pre-malignant senescent hepatocytes created a pro-tumorigenic and immunosuppressive environment[70,101]. CCL2, a cytokine, was identified as a key factor secreted by precancerous senescent hepatocytes, favouring the recruitment of CCR2+ immature myeloid cells (iMC). The differentiation and maturation of iMCs to macrophages is essential for precancerous senescent cell clearance. In contrast, iMC accumulation led to HCC through the inactivation of the NK cell function. Interestingly, Eggert et al. showed that tumour cells prevented the

maturation of iMC to macrophages through SASP secretion, which in turn resulted in tumour immune escape[70]. However, the pro- and anti-tumorigenic profiles of SASP are poorly defined and difficult to predict in the context of tumour relapse. How SASP modulates these opposing effects depending on the microenvironment or pathophysiological context remains to be determined. Moreover, it is not excluded that various senescent states might co-exist to shape the tumour microenvironment and modulate the pro- or anti-tumorigenic effects. Single-cell RNA sequencing could determine the senescent state phenotypes existing within a tumour.

## Strategies to eliminate dormant cells

**Strategies to target dormant quiescent cancer cells**. Because current therapies target proliferating tumour cells, an important question is which therapeutic approach would be best for eliminating dormant cancer cells. Here, we discuss three different strategies to target dormant quiescent cells (Fig. 2): "awakening" or enhancing the proliferation of dormant slow-cycling resistant cancer cells to increase their susceptibility to anti-proliferative drugs, keeping cells in a dormant state, and eradicating them while dormant-quiescent or slow-proliferating.

**"Awakening" of dormant quiescent cells**. Reactivating dormant quiescent cancer cells to make them rapidly re-enter the cell cycle is expected to improve their elimination by anti-proliferative drugs. When anti-proliferative chemotherapeutic agent resistant hematopoietic stem cells were pre-treated with IFNα, STAT1 and PKB/AKT were phosphorylated, increasing the expression of cell surface stem cell antigen-1, thereby inducing cell proliferation and efficient elimination by 5-fluorouracil (5-FU) in vivo[102]. In leukaemia, combined treatment with granulocyte colony-

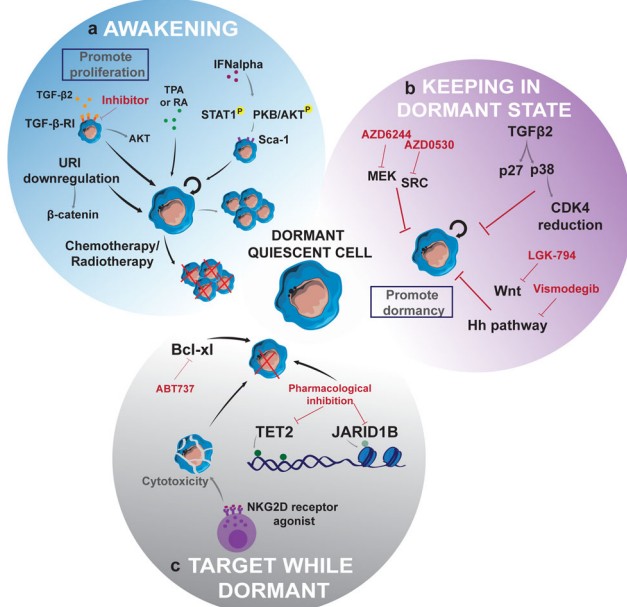

**Fig. 2 Schematic representation of the different strategies to eradicate dormant quiescent cancer cells.** Dormant quiescent or slow-cycling cells can mainly be targeted by three different strategies: **a** awakening, which aims to promote re-enter in cell cycle and proliferation of quiescent cells in order to be correctly targeted and eliminated by anti-proliferative therapies; **b** keeping the dormant state to avoid awakening and tumour relapse; and **c** targeting while dormant, which is based in targeting the crucial signalling pathways needed to keep cells in dormancy, such as epigenetic changes for example.

stimulating factor (G-CSF) and the cell-cycle dependant chemotherapeutic cytarabine enhanced the proliferation and elimination of quiescent stem cells in acute myeloid leukaemia mouse models[103]. However, G-CSF treatment followed by chemotherapies including cytarabine and mitoxantrone, or cytarabine, daunorubicin and thioguanine did not improve AML patients' outcome[104]. These results highlight the difficulties to translate findings from mice to humans, and point that strategies to awaken quiescent cells are not always easy to apply in patients.

The identification of essential pathways required to maintain a low proliferation rate or a dormant state could facilitate the design of effective "awakening" treatments that could be combined with anti-proliferative therapies to prevent cancer relapse. In line with this strategy, downregulation of the molecular chaperone URI (unconventional prefoldin RPB5 interactor) in intestinal label-retaining slow-cycling cells induced β-catenin expression and made cells highly proliferative and radiosensitive[105]. Reducing URI levels could thus be one way to increase the proliferation rate of slow-cycling cancer cells, making them more sensitive to chemo- or radiotherapy. Moreover, when proliferation of vismodegib-resistant quiescent cells was reinstated with retinoic acid or 12-O-tetradecanoylphorbol-13-acetate in basal cell carcinoma, remaining dormant cancer cells were completely eliminated by vismodegib, abolishing tumour relapse[21]. Therefore, reactivation of dormant cells into a proliferative stage followed by radio or chemotherapy could be an efficient therapeutic strategy against tumour relapse. Despite several studies suggesting that dormant cell reactivation as part of an "awakening" strategy could overcome chemotherapeutic drug resistance, the clinical implementation of this strategy is likely to be challenging because it is difficult to ensure that all cells will re-enter the cell cycle and then be eliminated. Indeed, recent studies suggest that such strategies could rapidly fuel tumour recurrence and worsen patient outcomes in some cases because of these remaining dormant quiescent cells. For instance, TGF-β2 was identified as a crucial inductor of dormancy in head and neck cancer cell lines. Inhibition of TGF-β receptor with LY-364947 in mouse models resulted in reactivation of dormant cells and an increase in metastatic burden in liver, spleen and bone marrow[106].

**Keeping cells in a quiescent state**. Another strategy to prevent tumour relapse involves maintaining dormancy to avoid rapid proliferation and tumour regrowth. Recent studies have revealed cues that promote cellular dormancy, which could enable the development of therapies that mimic the pro-dormancy mechanisms and thereby prevent tumour recurrence. Dormant cells are characterized by increased p38 MAPK and decreased ERK1/2 activities, which are widely used as dormancy markers[107]. Despite being active in senescent cells, strategies to modulate the p38/ERK pathways could lead to permanent growth arrest of quiescent cells, preventing tumour recurrence and metastasis[108–110]. Moreover, pharmacological inhibition of SRC and MEK could prevent the proliferative response of dormant quiescent cells to external stimuli and suppress their survival in breast cancer, preventing its recurrence[111]. Likewise, because dormant quiescent cancer cell awakening is thought to be the last step in metastatic outbreaks, blocking factors involved in this process could be a powerful and precise way of preventing metastasis[108]. In addition, activation of Wnt signalling was reported to be implicated in the switch from vismodegib-resistant quiescent cells to proliferative cancer cells, and a combination of inhibitors against Wnt and hedgehog pathways abolished relapse of basal cell carcinoma[21,44]. Several previous studies have proposed the crucial role of the extracellular matrix components in

quiescent cell awakening, particularly the β1 integrin signalling pathway[112,113]. Neutralizing antibody-mediated β1 integrin blockage leads to MLC phosphorylation, loss of actin stress fiber formation and prevents the switch of quiescent breast cancer cells to proliferative status[113]. Moreover, microenvironment-induced TGFβ2 signalling activates p27 and downregulates CDK4 via p38α/β, leading to cell dormancy[106] and activation of p38 induces p53 and BHLHB3 expression while inhibiting that of c-Jun and FoxM1[114].

Since the ability to switch from quiescence to proliferative state could be an issue for slow-cycling cells, keeping them in a quiescent state would be the best approach to prevent tumour recurrence. However, despite the aforementioned promising results, the proposed strategy requires the dormant state to be preserved for a long time to prevent tumour regrowth, which may be very difficult to achieve given the high adaptability of cancer cells to different scenarios. Furthermore, the strategies to maintain cells in dormancy for a long period requires a permanent treatment, which seems clinically unviable, mainly due to toxicity. Furthermore, long-term treatments could always give rise to resistance, causing more complexities in dealing with tumour relapse.

It is worth mentioning that Salvador-Barbero et al. suggest that treatment with CDK4/6 inhibitors to prevent cell cycle entry after treatment with antimitotic or DNA-damaging chemotherapeutics might improve pancreatic adenocarcinoma recovery[115]. Despite these attempts to target therapy-induced proliferative cancer cells, it remains to be seen in humans whether cell-cycle inhibitors can be used sequentially to efficiently target slow-proliferative cells. Moreover, it should be noted that some senescent cancer cells will remain in the tumour and could thus still rewire the microenvironment to promote recurrence.

**Targeting cells while quiescent**. Quiescent cancer cells have different characteristics than proliferating cells, opening alternative strategies to eradicate cancer cells in their dormant state. Insulin-like growth factor 1 (IGF-1)/IGF-1 receptor (IGF-1R) autocrine signalling and the subsequent AKT activation was identified as a common mechanism to promote dormancy in KRAS- and c-MYC null-pancreatic cancer cells; and pancreatic dormant cells were eliminated when treated with IGF-1R inhibitor[116]. In addition, quiescent slow-cycling cells display constant expression of Bcl-xl essential for their survival and inhibition of Bcl-xl by ABT-737 resulted in the elimination of quiescent slow-cycling non-small cell lung cancer cells, highlighting the potential therapeutic use of ABT-737 to eradicate slow-cycling cells[117]. Likewise, quiescent persistent cancer cells were shown to be sensitive to ferroptosis, a programmed cell death induced by lipid peroxides accumulation. The phospholipid glutathione peroxidase GPX4 protects against membrane lipid peroxidation and in turn, prevents ferroptotic cell death[118]. GPX4 inhibitor RSL3 selectively reduced the residual persistent cell pool in several types of cancer cell lines including melanoma (A375 cell line), breast (BT474 cell line), lung (PC9 cell line) and ovarian (Kuramochi) cancer cells as well as in A375 melanoma cell lines-derived xenograft models[119]. Persistent quiescent cells in colorectal cancers could also be eliminated by targeting autophagy. Quiescent cells treated with inhibitors against ULK1, a crucial kinase activating autophagy, in combination with a standard chemotherapy treatment (CPT-11), failed to regrow and underwent apoptosis, even after treatment discontinuation[24]. Moreover, since quiescent cancer cells evades NK cell recognition by downregulating their stress ligand ULBP1, the use of specific agonists for NKG2D receptor-activating NK cells could lead to the destruction of quiescent cells[43].

Unfortunately, the induction of cellular dormancy and retardation of the rate of proliferation appear to be complex processes that may involve robust epigenetic reprogramming, and little is currently known about the epigenetics of slow-cycling cells. The epigenetic enzyme TET2 may be a key factor controlling the numbers and survival of slow-cycling cancer cells as well as tumour recurrence. 5-hydroxymethylcytosine generated by the activity of TET2 was identified as a predictive biomarker of relapse and survival in cancer patients, suggesting that TET2 could be a potential drug target for slow-cycling cell elimination[2]. In addition, in vitro experiments on melanoma cells showed that both cytotoxic and targeted cancer chemotherapeutic agents caused uniform enrichment of cells expressing the H3K4 demethylase JARID1B. It was therefore postulated that targeting the slow-cycling cell population by inhibiting this enzyme's demethylase activity while simultaneously applying conventional anti-proliferative therapy could help eradicate all melanoma cells[120].

A strategy based on targeting dormant cells would have to be efficient enough to ensure that no slow-cycling/dormant quiescent cells remain. Since no diagnostic tools currently exist to detect dormant quiescent cells in patients, such efficiency will probably be difficult to achieve. Nevertheless, the evidence accumulated to date strongly suggests that persistent or untargeted slow-cycling quiescent cells can become more aggressive and lead to worse prognoses. Identifying unique features and markers of quiescent cells could also allow the development of strategies directing the immune system against dormant cells. As proposed for senescent cells[121], developing chimeric antigen receptor (CAR) T cells could be useful to directly recognize and eliminate quiescent dormant cancer cells. These innovative strategies stress out the urgent need to discover surface markers of slow-cycling or quiescent cancer cells. Detecting dormant cells through specific labelling in vivo would help in this task.

**Strategies to eradicate senescent cells**. As noted above, the SASP can control surrounding cells via paracrine loops. However, because surrounding cells can act as signal relays, it can also indirectly influence the SASP-displaying senescent cells themselves[95]. Due to their persistent SASP secretion, these cells will be surrounded by radioprotective and chemoprotective factors, as well as growth and angiogenic factors that support tumour progression. It is also known that as the ratio of senescent cells to immune cells increases, senescent cells become more tumour-promoting rather than tumour-suppressive[122].

Owing to their molecular complexity and interactions, different strategies to eradicate senescent cells have been proposed (Fig. 3). As senescent cancer cell activity is mainly directed via SASP secretion, SASP modulation for therapeutic purposes could be a promising way of preventing tumour relapse, also known as senomorphic therapy. Various inhibitors have been proposed to induce a switch from pro-tumorigenic SASP to tumour-suppressive SASP. The secretome of senescent cells relies on their genetic background. For instance, senescent cells present in the PTEN-null prostate tumours revealed an immunosuppressive SASP mainly controlled by NF-kB and STAT3 signalling. Treatment of these cells with JAK2/STAT3 inhibitors provoked a SASP secretory switch resulting in an anti-tumorigenic secretome-activating an immunosurveillance response[123].

Another strategy would be to target senescent cells by blocking the paracrine effects of SASP. Laberge et al. showed that senescence signal-mediated IL-1α translation was mTOR-dependent, and thus rapamycin sensitive. Rapamycin treatment leads to the blockage of IL-1α translation and in turn, reduced NF-κB-mediated SASP factors gene expression activated downstream of

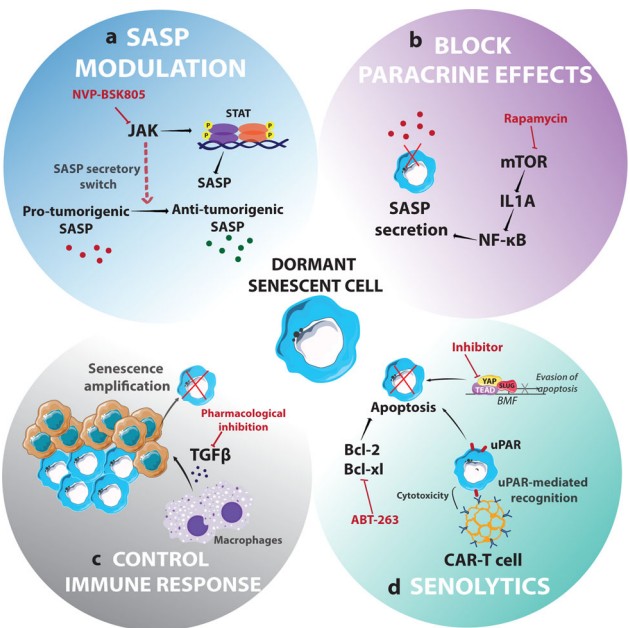

**Fig. 3 Schematic representation of the different strategies to eradicate dormant senescent cancer cells.** Strategies to eradicate dormant senescent cancer cells have been classified in four groups: since senescent cancer cell activity is mainly directed via SASP secretion, two different approaches can be done **a** SASP modulation; **b** blocking the paracrine effects of SASP; **c** control the immune response in order to avoid senescence amplification; **d** use senolytic compounds that directly target senescent cells.

IL1R[124]. Thus, rapamycin suppressed both the establishment and maintenance of SASP, suggesting that rapamycin is a potential viable therapeutic approach to target senescent cells. However, since rapamycin treatment is not exclusive for senescent cells, this therapy might also affect healthy epithelial cells and have deleterious effects[125,126]. Another example is a study which identified the critical role of the rasGAP SH3-binding protein 1 (G3BP1) in SASP secretion[127]. G3BP1 depletion in primary human lung fibroblasts induced a "SASPless" phenotype of senescent cells, which were unable to promote tumour growth in vitro and in vivo. Thus, G3BP1 inhibition could block the paracrine effects of senescent cells and in turn, its pro-tumorigenic effect.

Controlling the immune responses to SASP would be an alternative immunotherapeutic approach to ameliorate or promote the anti-tumour activity of crosstalk between SASP and immune cells. SASP-mediated macrophage recruitment leads to macrophage-dependent paracrine TGFβ signalling, which induces senescence amplification in liver injury models. This mechanism could thus be exploited to target TGFβ signalling and thereby reduce the non-cellular autonomous effects of senescence on tumorigenesis[128].

Small-molecule agents known as senolytics that selectively target and eliminate dormant senescent cells are becoming increasingly attractive as options for treating cancer and preventing relapse[129–131]. ABT263, also known as Navitoclax, a well-known Bcl-2 and Bcl-xl anti-apoptotic inhibitor selectively targets and eliminates senescent cells by inducing apoptosis[132]. Furthermore, a senescent-like dormant phenotype was observed following EGFR/MEK combinatorial treatment in non-small cell lung cancer that enabled tumour recurrence. The authors indicated that YAP/TEAD-mediated epigenetic alterations, via SLUG, a transcription factor of EMT process, suppressed the pro-

apoptotic factor BMF, leading to survival of cancer cells[133]. As YAP/TAZ inactivation had no relevant side effects on the basal homeostasis of surrounding healthy adult tissue[134], they proposed that pharmacological inhibition of YAP/TEAD could lead to apoptosis of senescent-like cancer cells, resulting in tumour regression. Very recently, Amor et al. suggested the therapeutic use of CAR T cells as senolytics to target senescent cells. The authors identified urokinase-type plasminogen activator receptor (uPAR) as a cell-surface protein that is induced during senescence and demonstrated that uPAR-specific CAR T cells efficiently depleted senescent cells in vitro and in various disease settings[121]. Senolytics could be used in combination or sequentially with chemotherapeutic drugs or radiotherapy[135]. This two-hit anti-cancer strategy would involve first inducing senescence with chemotherapeutic agents and then eliminating senescent cancer cells by directly targeting them. However, this would require the development of biomarkers to classify cells exhibiting TIS as either tumour-suppressive or pro-tumorigenic[90].

Interestingly, an elegant inducible genetic system to eliminate in mice the p16[INK4a]-positive, senescent cells demonstrated that such elimination of senescent cells delayed aged-related disorders[80,136]. Thus, therapeutic elimination of senescent cells could be a good approach to delay and/or to treat age-related diseases, including the pro-tumorigenic effects of senescent cells.

Another therapeutic strategy that has been considered is to promote homogeneous senescence within the tumour. A brief exposure to Palbociclib via lysosomal trapping selectively inhibits CDK4/6, resulting in stable cell-cycle arrest and long-term senescence[137]. Moreover, based on CRISPR-mediated genetic and chemical screens, it was proposed that suppressing the SWI/SNF component SMARCB1 induces senescence in melanoma by strongly activating the MAP kinase pathway[138]. However, inducing homogenous senescence could be challenging, and senescent cells could rewire the tumour microenvironment in ways that would promote tumour relapse, potentially making this strategy more harmful than helpful in some cases. Furthermore, a small fraction of senescent cells could escape from their dormant state by senescence-associated stemness, and hence promote tumour growth potential[30]. Therefore, pharmacological strategies aimed to eliminate senescent cells before a fraction of them implement features of senescence-associated stemness and re-enter cell cycle would certainly avoid tumour relapse.

## Concluding remarks

Tumour relapse is a complex and poorly defined phenomenon that limits our ability to completely cure cancer. Several studies have highlighted the presence of slow-cycling or slow-proliferating cancer cells and senescent cancer cell populations in tumours, neither of which are targeted by common cancer treatments, such as chemo and radiotherapy. These persistent cancer cell population is residual and undetectable and might be the cells at the origin of tumour relapse after several months or even years and once the treatment is stopped. Likewise, TIS could also induce dormancy through the appearance of senescent cells. Quiescent cells are supposed to contribute to tumour relapse by re-entering the cell cycle most likely due to appropriate fine-tuned microenvironment, while senescent cells may reinforce tumour regrowth though SASP and immune system modulation. The crosstalk between slow-cycling and senescent cells is not excluded and SASP secretion could fuel the proliferation of slow-cycling cells. SASP could also induce stemness of surrounding cells arguing to the pro- and anti-tumorigenic roles of senescent cells. However, much remains to be learned about the exact role of dormant cells in tumour recurrence. To this end, there is a clear need for new biomarkers and genetically engineered mouse

models that can be used to label, track and monitor dormant cells after chemo- or radiotherapy to clarify their roles and functions during tumour relapse. This in turn may facilitate the design of new drugs targeting dormant cells and guide the development of new therapies to prevent the potentially fatal recurrence of tumours in cancer survivors.

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

## Acknowledgements

This work was funded by the State Research Agency (AEI, 10.13039/501100011033) from the Spanish Ministry of Science and Innovation (projects granted to N.D. SAF2016-76598-R, SAF2017-92733-EXP, RTI2018-094834-B-I00 and RED2018-102723-T), cofounded by European Regional Development Fund (ERDF). K.S.D.F. is recipient of a fellowship from the AECC Scientific Foundation (Madrid). This work was developed at the CNIO funded by the Health Institute Carlos III (ISCIII) and the Spanish Ministry of Science and Innovation.

## Author contributions

K.S.D.F. and N.D. wrote the manuscript.

## Competing interests

The authors declare no competing interests.
