## [Peer Review File · Communications Biology]

Reviewers' comments:

Reviewer #1 (Remarks to the Author):

Santos-de-Frutos-K and Djouder-N, When Dormancy Fuels Tumour Relapse
Submitted to Communications Biology

In this review manuscript, the two authors aim at determining contributions of slowly and non-dividing cancer cells on tumor recurrence, thereby pinpointing differences of dormant-quiescent and dormant-senescent cells. In the course of this extensive manuscript, the outlined distinctions seem to vanish, leaving the reader with more uncertainty what is actually talked about. This is only partly due to structure and precise semantics in writing, it is certainly due to biological conditions that are less clearly defined as being distinct. The authors should find an optimized way to handle this intrinsic problem with enhanced readability. The therapeutic perspectives as presented (awakening, keeping in dormancy, targeting in dormancy), although not exhaustive, are interesting, but scratch only at the surface of the consecutive effects, for instance on the immune system, all these interventions may have. Moreover, the authors – both natural scientists – seem to lose faith in their suggestions when considering how long and how efficient some of the therapeutic concepts one would need to apply to see their hoped-for efficacy unfolding. Perhaps, a critical view by a clinically experienced oncologist might help to ground this part in medical practice.

In general, this is an interesting review on a timely topic, albeit not yet ready for publication. Although quite comprehensive, some key references have been missed and should be added to ensure a balanced view on the topic (or to give credit to inaugural, not follow-up papers). For details, see my comments below.

Major concerns and comments

1. While the assumption of non- or slow-cycling dormant cancer cells as the source of relapse might be true, the authors have not convincingly explained why, as the obvious alternative, not small fractions of rapidly dividing tumor cells survive conventional therapies (since those may simply fail to reach out to every given cell, and certain subclones in heterogenous tumor cell populations, independent of dormancy, might possess apoptosis-countering mutations, thereby resisting drug-induced apoptosis). This view particularly applies to cancer stem cells, which may possess stem-typical mechanisms (such as ABC transporters) that renders them susceptible to conventional chemotherapeutics. Those small survivor fractions might very well consist of dividing tumor cells – if they are just small enough (reflecting a reduction in tumor burden by the several orders of magnitude), it will take quite some time until a relapse may emerge clinically.

2. The dormancy concept is certainly an appealing explanation of tumor types known to frequently present with metastasis a decade or later after initial tumor diagnosis. However, the authors should state that such behavior is rather the exception, not the rule. Most cancer types recur within the first few years, not later.

3. Line 92: Whether senescence actually represents a truly irreversible arrest condition, is currently under intense debate. Despite robustly fulfilling typical features of senescence, senescent cells may, if senescence-mandatory maintenance genes are no longer expressed, resume proliferation. Hence, cells that underwent cellular senescence can become, occasionally, post-senescent. This is an important notion the authors should conceptually address, since distinctions to quiescence as a reversible state may vanish – or actual cell biological features acquired in senescence and further propagated in cells after senescence might become critically important for their distinctly more aggressive behavior.

4. Line 124: It is an important point to consider that slowly dividing or even arrested cells may acquire further mutations. However, it is not clear at this point of the review, why quiescent cells need new mutations to escape the immune system. Are the authors implying a general anti-tumor immunosurveillance to apply, or referring to dormancy-associated immunogenic changes?

5. Figure 1: The authors imply that secreted factors from senescent cells (so called SASP) may impinge on neighboring cells to modulate their stemness capacity. A key publication in this regard has been missed (Mosteiro-L et al., Science, 2016). Importantly, such Figure implies a key role for senomorphic, SASP-blunting therapies. This, however, is not part of therapeutic perspectives provided by this review. On the contrary, cell-autonomous reprogramming of senescent cells that may occasionally resume proliferation (see Milanovic-M et al., Nature, 2018), hence underscoring the need to eliminate those cells before they “wake up”, has not been addressed either. Moreover, SASP factors might not only create a pro-tumorigenic/mitogenic, inflammatory/immune-suppressive environment, but induce a secondary, paracrine form of cellular senescence, thus, can have tumor-suppressive potential – an aspect underrecognized in this manuscript (see Acosta-J et al., Nat Cell Biol, 2013).

6. Line 281: Obviously, the fate of senescent cells is complex. Certainly, it is not true that “Unlike in apoptosis, cells that enter senescence are not eliminated”. There is accumulating evidence that cells of both the innate and the adaptive immune system eliminate senescent cells (see Xue-W, Nature, 2007; Kang-TW et al., Nature, 2011; Reimann-M et al., Blood, 2020). The maintained or lost endogenous clearance capacity towards senescent cancer cells is a key determinant of long-term tumor fate, and deserves higher visibility in text and figures of this review.

7. Line 383: Beyond the ethically highly problematic side of “actively awakening dormant cancer cells” in patients to – hopefully – kill them all afterwards, instead of rephrasing the strategy to “preventing cancer cells to become dormant during induction therapy”, the statement on G-CSF in leukemia is misleading, since phase III trials have shown that the addition of G-CSF to a classic, AraC-containing “7 + 3” induction regimen did not improve outcome of AML patients (see Krug-U et al., Leukemia, 2016). Are the authors implying such strategy might work in post-induction patients in clinical complete remission (again, ethically highly problematic...), or when facing an overt relapse? If the latter is the case, would an “awakening” strategy suffice in their view, without a debulking re-induction therapy?

8. Line 439: The authors themselves raise doubts that their suggested strategy, to keep cancer cells in a dormant state, might be clinically feasible. Before giving up so easily, immunological aspects, i.e. help by the immune system to clear those dormancy-enforced cells, might be discussed in greater detail.

9. It might help to seek a counseling opinion on treatment perspectives, their assumed feasibility and expected strength by a clinical colleague, namely a patient-caring oncologist.

Minor concerns

1. The manuscript should be seen by a native speaker

2. Line 34: “Most treated patients relapse after surgery or adjuvant therapies” is simply not true

3. Line 63: The class of targeted therapeutics, i.e. signaling inhibitors such as TKIs, but also proteasome blockers, HDAC inhibitors and many other biologicals were missed here.

4. Line 65: “Cytotoxic” agents kill cells, “Genotoxic” agents provide DNA damage.

5. Line 81: What do the authors mean by “therapy-induced breast cancer relapse”?
6. Line 104: “Quiescence is a cellular process that preserves stem cell function in case it is needed in tissue homeostasis or repair” – actually applies similarly to senescence (see Demaria-M et al., Dev Cell, 2014).
7. Line 128: The authors refer to NKG2D ligands to activate NK cells (not NKT cells...) in the context of quiescence. It should be noted that such mechanism reportedly applies to senescent cells as well (see Iannello-A et al., J Exp Med, 2013)
8. Line 135: replace “authors fail to demonstrate” by “authors did not show that”
9. Line 164: “Senescent cells are irreversibly arrested in the G1-G1/M phase” – don’t understand, what cell-cycle phase are the authors referring to? Classic cellular senescence is a lasting G1-phase arrest, mediated by an epigenetically locked Rb/E2F machinery (see Serrano-M et al., Cell, 1997, and Narita-M et al., Cell, 2003)
10. Line 247: The link between “elimination of senescent cells improves these pathologies [such as liver fibrosis]” and “Krizhanovsky showed that hepatic stellate cells... ..undergo senescence... ..enhancing the expression of the matrix metalloproteases with fibrolytic activity... ..hence limiting liver fibrosis” seems to argue for the opposite...
11. Line 428: As above, p38MAPK is not an exclusive feature of dormant cells, but a central signaling cascade active in senescent cells (see Freund-A et al., EMBO J, 2011) – in other words, the distinction of dormant, quiescent and senescent cells remains blurry, potentially for the reason that these terms may actually describe largely similar conditions. Table 1 is not sufficient to elucidate the problem – it’s rather the missed aspects (e.g. genomic re-organization, alterations of the nuclear envelope, the expanded lysosomal compartment a.o.), which might help to distinguish and to conclude structure-to-function implications if those cells re-enter the cell-cycle.

Reviewer #2 (Remarks to the Author):

In this manuscript Santos-de-Frutos et al review the current status of research in the area of tumor dormancy. This is a very thoroughly researched, well-written and timely review focusing on the role of therapy, senescence and therapy induced senescence in tumor dormancy. While both therapy and senescence have been postulated to have a potential role in the induction/maintenance/exit from dormancy, proof-of concept studies in these areas are lacking. This review provides a comprehensive overview of the current status of ongoing research in this field and will influence thinking in this field. However, there are a few minor concerns

There are few grammatical errors and typos that needs fixing.

The abstract needs to focus more on the role of senescence and therapy in the dormancy program rather than a generalized abstract about dormancy.

Although the review focuses on the role of senescence in dormancy program, the additional sections discussing the anti- and pro-tumorigenic roles of senescence distracts away from the emphasis on its role in dormancy. While there are plenty of reviews examining the role of senescence in the former, not many discuss the role of this important physiological program in regulating tumor dormancy. The authors therefore need to reconsider the addition of these sections to the manuscript.

Detailed responses to the Reviewers' comments

REVIEWER #1

Santos-de-Frutos-K and Djouder-N, When Dormancy Fuels Tumour Relapse Submitted to Communications Biology. In this review manuscript, the two authors aim at determining contributions of slowly and non-dividing cancer cells on tumor recurrence, thereby pinpointing differences of dormant-quiescent and dormant-senescent cells. In the course of this extensive manuscript, the outlined distinctions seem to vanish, leaving the reader with more uncertainty what is actually talked about. This is only partly due to structure and precise semantics in writing, it is certainly due to biological conditions that are less clearly defined as being distinct. The authors should find an optimized way to handle this intrinsic problem with enhanced readability. The therapeutic perspectives as presented (awakening, keeping in dormancy, targeting in dormancy), although not exhaustive, are interesting, but scratch only at the surface of the consecutive effects, for instance on the immune system, all these interventions may have.

Moreover, the authors “ both natural scientists “ seem to lose faith in their suggestions when considering how long and how efficient some of the therapeutic concepts one would need to apply to see their hoped-for efficacy unfolding. Perhaps, a critical view by a clinically experienced oncologist might help to ground this part in medical practice.

In general, this is an interesting review on a timely topic, albeit not yet ready for publication. Although quite comprehensive, some key references have been missed and should be added to ensure a balanced view on the topic (or to give credit to inaugural, not follow-up papers). For details, see my comments below.

We are very grateful to Reviewer#1 for their general interest on our manuscript and for finding it interesting, timely and appropriate for Communications Biology. We are also thankful to this Reviewer for dedicating part of their time to review this manuscript and for raising critical points aiming to improve this paper.

Changes in the manuscript are highlighted in blue.

Major concerns and comments

1. While the assumption of non- or slow-cycling dormant cancer cells as the source of relapse might be true, the authors have not convincingly explained why, as the obvious alternative, not small fractions of rapidly dividing tumour cells survive conventional therapies (since those may simply fail to reach out to every given cell, and certain subclones in heterogeneous tumour cell populations, independent of dormancy, might possess apoptosis-counteracting mutations, thereby resisting drug-induced apoptosis). This view particularly applies to cancer stem cells, which may possess stem-typical mechanisms (such as ABC transporters) that renders them susceptible to conventional chemotherapeutics. Those small survivor fractions might very well consist of dividing tumor cells “ if they are just small enough (reflecting a reduction in tumor burden by the several orders of magnitude), it will take quite some time until a relapse may emerge clinically.

This is an interesting point raised by Reviewer #1, which definitely needs a worthy clarification.

Although dormant cells are thought to be involved in tumour relapse, other alternatives do exist. Cancer stem cells (CSCs) are defined by their tumour initiating ability, capacity to self-renew and ability to differentiate into various tumour bulk subpopulations and several studies have suggested their requirement in tumour relapse (Clarke et al, Cancer Res 2006). It needs to be noted that there are several similarities between the concept of cancer dormancy and the CSC theory of tumour development (Aguirre-Ghiso, Nat Rev Cancer 2007). CSCs, like dormant cancer cells, survive conventional cancer therapies and can evade antitumour immune responses. The general definition of CSC is the ability to initiate tumour outgrowth. However, several evidences suggest that CSCs can consist of heterogeneous subpopulations, including fast-cycling and slow-cycling or quiescent CSCs (Agudo et al, Immunity 2018). This late subpopulation could be directly linked to dormant cancer cells and might therefore exploit dormancy states to ensure long-term tumour maintenance upon different environments. Thus, CSCs could be considered as quiescent subpopulations critical in the switch from dormancy to proliferation state to promote tumour outgrowth. Hence, insights in CSC biology could help in proceed in the understanding of the mechanisms underlying cancer dormancy.

This part has been discussed in **lines 62-80** of the revised manuscript.

2. The dormancy concept is certainly an appealing explanation of tumor types known to frequently present with metastasis a decade or later after initial tumor diagnosis. However, the authors should state that such behaviour is rather the exception, not the rule. Most cancer types recur within the first few years, not later.

We thank Reviewer #1 for pointing out to the concept of the time needed to the tumour relapse. Indeed, as this Reviewer has pointed out, although there is no strict definition regarding the time for disease recurrence, it is generally thought that dormancy can persist in a latent state from months to several years until tumour regrowth in the primary site or metastasis occurs.

This part has been discussed in **lines 102-107** of the revised manuscript.

3. Line 92: Whether senescence actually represents a truly irreversible arrest condition, is currently under intense debate. Despite robustly fulfilling typical features of senescence, senescent cells may, if senescence-mandatory maintenance genes are no longer expressed, resume proliferation. Hence, cells that underwent cellular senescence can become, occasionally, post-senescent. This is an important notion the authors should conceptually address, since distinctions to quiescence as a reversible state may vanish “ or actual cell biological features acquired in senescence and further propagated in cells after senescence might become critically important for their distinctly more aggressive behaviour.

Reviewer #1 is totally right with this comment, and we thank them for raising it. Even though senescence is considered a completely irreversible growth arrest, more and more studies are currently pointing out to the fact that some senescent cells could escape senescence state and re-enter the cell cycle.

As mentioned by Reviewer #1, and as Milanovic *et al* suggested, we could have other cells types known as “post-senescent” cells that could maintain some senescence-associated aggressive features, as well as senescence-associated stemness that could enforce tumour relapse (Milanovic et al, Nature 2018). As mentioned in the section “strategies to eliminate dormant senescent cells”, strategies to eliminate senescent cells before some of them escape the arrested condition and become post-senescent cells are needed. However, proofs of concepts studies are needed to show how aggressive can be those post-senescent cells. Yet, different senescent state might exist and this cannot be excluded.

This part has been discussed in **lines 269-278** and **lines 679-683** of the revised manuscript.

4. Line 124: It is an important point to consider that slowly dividing or even arrested cells may acquire further mutations. However, it is not clear at this point of the review, why quiescent cells need new mutations to escape the immune system. Are the authors implying a general anti-tumor immunosurveillance to apply, or referring to dormancy-associated immunogenic changes?

More and more studies have proposed that quiescent cancer cells are able to evade immune system. Different reasons could be involved in immunosurveillance evasion: i) acquisition of new mutations that might facilitate quiescent cancer cells to escape immune system and/or ii) the loss of major neoantigens of cancer cells. Both cases are thought to be quiescent-state associated immunogenic changes that can facilitate immunosurveillance evasion.

Agudo *et al* and Malladi *et al* demonstrated that immune protection is a slow-cycling cell property. This could explain the ability of quiescent cancer cells to promote tumour regrowth as well as metastasis (Agudo *et al*, Immunity 2018; Malladi *et al*, Cell 2016).

As suggested by this Reviewer, we have revised this section and re-discussed it in **lines 167-182** of the revised manuscript.

5. Figure 1: The authors imply that secreted factors from senescent cells (so called SASP) may impinge on neighbouring cells to modulate their stemness capacity. A key publication in this regard has been missed (Mosteiro-L et al., Science, 2016). Importantly, such Figure implies a key role for senomorphic, SASP-blunting therapies. This, however, is not part of therapeutic perspectives provided by this review. On the contrary, cell-autonomous reprogramming of senescent cells that may occasionally resume proliferation (see Milanovic-M et al., Nature, 2018), hence underscoring the need to eliminate those cells before they “wake up”, has not been addressed either. Moreover, SASP factors might not only create a pro-tumorigenic/mitogenic, inflammatory/immune-suppressive environment, but induce a secondary, paracrine form of cellular senescence, thus, can have tumour-suppressive potential “ an aspect underrecognized in this manuscript (see Acosta-J et al., Nat Cell Biol, 2013).

We are grateful to Reviewer #1 for suggesting important publications on the topic of SASP paracrine effects. As pointed out, studies by Mosteiro *et al* imply that senescent cells are able to promote reprogramming in neighbouring cells mainly by IL-6 secretion (Mosteiro et al, Science 2016) (**lines 314-316**).

Regarding senomorphics, they are molecules that can interfere with senescent cells in an indirect way by affecting SASP secretion. Since senescent cell activity is mainly driven by SASP secretion, targeting those cells by targeting SASP levels would be important. In fact, as we have already mentioned (**line 613**), SASP modulation for therapeutic purposes could be a promising way of preventing senescence-mediated tumour relapse.

According to the study by Milanovic *et al*, we have highlighted the importance of eliminating senescent cells as soon as possible, before a fraction of them reverses the arrest state and promotes tumour outgrowth (Milanovic et al, Nature 2018) (**lines 679-683**).

Concerning the tumour suppressive role of the paracrine senescence, as mentioned in the manuscript, SASP secretion mainly consists of two phases: “first wave” that induces paracrine senescence on the surrounding cells and “second wave” that induces the secretion of inflammatory cytokines that in turn favour immune cell infiltration and elimination of senescent cells. Several studies have been mentioned that have demonstrated the tumour suppressive effect of senescent cells-mediated paracrine function (**lines 394-402**).

6. Line 281: Obviously, the fate of senescent cells is complex. Certainly, it is not true that “Unlike in apoptosis, cells that enter senescence are not eliminated”. There is accumulating evidence that cells of both the innate and the adaptive immune system eliminate senescent cells (see Xue-W, Nature, 2007; Kang-TW et al., Nature, 2011; Reimann-M et al., Blood, 2020). The maintained or lost endogenous clearance capacity towards senescent cancer cells is a key determinant of long-term tumor fate, and deserves higher visibility in text and figures of this review.

We agree with the comment raised by Reviewer #1. With the sentence mentioned (“*Unlike in apoptosis, cells that enter senescence are not eliminated*”) we meant to say that even though apoptosis and senescence might have high levels of DNA damage, senescence does not lead specifically to cell death; instead, those cells are able to remain metabolically active.

As pointed out by Reviewer #1, immune cells from innate and immune system are able to promote the clearance of senescent cells (as mentioned in the section “Anti-tumorigenic effects of senescent cells”).

We agree that this is an awkward sentence that has been corrected in the revised manuscript. Moreover, the importance of the immune system cells in senescence clearance has been better discussed in **lines 396-422** of the revised manuscript. Furthermore, **Figure 1** has been corrected according to Reviewer #1’s suggestions.

7. Line 383: Beyond the ethically highly problematic side of “actively awakening dormant cancer cells” in patients to “hopefully “ kill them all afterwards, instead of rephrasing the strategy to “preventing cancer cells to become dormant during induction therapy”, the statement on G-CSF in leukemia is misleading, since phase III trials have shown that the addition of G-CSF to a classic, AraC-containing “7 + 3” induction regimen did not improve outcome of AML patients (see Krug-U et al., Leukemia, 2016). Are the authors implying such strategy might work in post-induction patients in clinical complete remission (again, ethically highly

problematic”), or when facing an overt relapse? If the latter is the case, would an “awakening” strategy suffice in their view, without a debulking re-induction therapy?

We are very thankful to Reviewer #1 for helping us to improve this manuscript and better discuss the clinical evidences and relevance of the mouse work.

As noted by Reviewer #1, the study by Saito *et al* suggest the improvement of G-CSF to eliminate quiescent stem cells and thereby increased survival rates in acute myeloid leukemia (AML) mouse model (Saito et al, Nat Biotechnol 2010). However, when these results were translated to AML patients, G-CSF treatment before induction therapies such as cytarabine and mitoxabtrone or cytarabine, daunorubicin and thioguanine did not improve AML patients’ outcome.

If the strategy is based on awakening therapies followed by chemotherapeutic therapies, it would have, in theory, facilitated quiescent cancer cells to become sensitive to chemotherapy. However, in the case of G-CSF clinical trials performed by Krug *et al* have shown that G-CSF treatment does not promote quiescent cancer cell elimination and in turn, survival improvement (Krug et al, Leukemia 2016). Regarding the point raised by Reviewer #1, we do not think that the strategy of awakening would have a benefit when facing an overt relapse, since at that moment cells that were quiescent would have probably enter the cell cycle and thus, awakening them would not be necessary.

We would like to emphasize that as mentioned in the manuscript, despite several studies have suggested that the awakening of quiescent cancer cells could overcome chemotherapeutic resistance and reduce tumour outgrowth, the clinical implementation of this strategy is still challenging and can induce an opposite effect than expected, an uncontrolled proliferation of post-quiescent cell which might result in metastasis.

Nevertheless, we found interesting and essential to present all the possible and proposed strategies to eradicate quiescent cancer cells. This part has been better discussed in **lines 473-480** of the revised manuscript.

8. Line 439: The authors themselves raise doubts that their suggested strategy, to keep cancer cells in a dormant state, might be clinically feasible. Before giving up so easily, immunological aspects, i.e. help by the immune system to clear those dormancy-enforced cells, might be discussed in greater detail.

Our idea has been to summarize several studies published to date the possible strategies to target quiescent cancer cells, always pointing out the positive and negative aspects of each of them. Of note, since the main problem of slow-cycling cells is their ability to switch to a proliferative state and thus promote tumour growth, at the first glance, keeping them in a dormant state would be the best approach to prevent tumour relapse. However, as pointed out in the manuscript, maintaining those cells in a non-proliferative state would require long-term treatments which, for our knowledge, seem quite unfeasible due to toxicity, time and many other constraints for the patient. We have therefore emphasized that targeting them while quiescent (or dormant) could be the best strategies to follow, but of course, deeper understanding of quiescent cell biology is needed.

We have revised this section and re-discussed it as suggested by this Reviewer in **lines 524-528 and 593-599** of the revised manuscript.

9. It might help to seek a counselling opinion on treatment perspectives, their assumed feasibility and expected strength by a clinical colleague, namely a patient-caring oncologist.

We thank Reviewer #1 for their suggestion; however, we consider that treatment perspectives from an oncologist's point of view would be out of the scope of this review, and might require a complete perspective/review on this topic. We have therefore decided to focus on the current review not to further extend over our lines of discussions. We hope that this Reviewer agrees with us.

Minor concerns

1. The manuscript should be seen by a native speaker

We apologize for the grammatical mistakes that could have been found by Reviewer #1. Following their suggestion, the manuscript has been edited by a native speaker.

2. Line 34: “Most treated patients relapse after surgery or adjuvant therapies” is simply not true

Reviewer #1 is right. This is a clumsy sentence.

Instead of saying “most treated patients” we would better say “a high percentage of treated patients”. In fact, relapse statistics vary widely between cancer types as well as the stages. In the case of glioblastoma, as mentioned in the manuscript, recurrence is almost inevitable. While 20-40% of breast cancer patients suffer relapse over a 5-year period.

This has been now corrected in the revised manuscript.

2. Line 63: The class of targeted therapeutics, i.e. signaling inhibitors such as TKIs, but also proteasome blockers, HDAC inhibitors and many other biologicals were missed here.

We are thankful for Reviewer #1's suggestions. We have now discussed and cited the most commonly used targeted therapies.

3. Line 65: “Cytotoxic” agents kill cells, “Genotoxic” agents provide DNA damage.

As rightfully pointed out by Reviewer #1, cytotoxic agents are compounds that kill cells, including cancer cells, and genotoxic agents produce DNA lesions or directly damage it. In agreement with this, the sentence has been corrected.

5. Line 81: What do the authors mean by “therapy-induced breast cancer relapse??

We apologize if the idea was not correctly explained. We wanted to point out that studies done in breast cancer cell lines mimicking aromatase-induced resistance have identified a pre-adapted cell population which triggers a dormant state and facilitates breast cancer relapse. The sentence has already been corrected in the manuscript in order to clarify it.

6. Line 104: “Quiescence is a cellular process that preserves stem cell function in case it is needed in tissue homeostasis or repair” “ Actually applies similarly to senescence (see Demaria-M et al., Dev Cell, 2014).

As pointed out by Reviewer #1, stem cell features are not limited to dormant quiescent cells. Indeed, Milanovic *et al* demonstrated that adult stem cell signature is strongly enriched in senescent cells, and thus, share some similarities with slow-cycling cells (Milanovic et al, Nature 2018). In fact, the so called senescence-associated stemness (SAS) was confirmed in several senescence models such as replicative senescence and stress-induced senescence in both human and mouse cells.

This has been corrected in the manuscript and discussed on **lines 145-146 and 269-274** of the manuscript.

7. Line 128: The authors refer to NKG2D ligands to activate NK cells (not NKT cells”) in the context of quiescence. It should be noted that such mechanism reportedly applies to senescent cells as well (see Iannello-A et al., J Exp Med, 2013)

We are grateful to Reviewer #1 for pointing out the important role of NKG2D mediated-cancer cell elimination not only for quiescent cancer cells but also for senescent cancer cells.

NK cells are also implicated in senescent cancer cell clearance. As shown by Iannelo et al, p53-mediated senescence induced several chemokine and cytokine secretion that in turn activated NK recruitment into the tumour and the expression of NKG2D ligands in cancer cells, which resulted in senescent cancer cell elimination (Iannello et al., J Exp Med, 2013).

This has been discussed on **lines 182-185** of the manuscript.

8. Line 135: replace “authors fail to demonstrate” by “authors did not show that”

We thank Reviewer #1 for the suggestion. We have corrected it in the manuscript.

9. Line 164: “Senescent cells are irreversibly arrested in the G1-G1/M phase” “ don”t understand, what cell-cycle phase are the authors referring to? Classic cellular senescence is a lasting G1-phase arrest, mediated by an epigenetically locked Rb/E2F machinery (see Serrano-M et al., Cell, 1997, and Narita-M et al., Cell, 2003).

We apologize for this typo mistake.

As Reviewer #1 mentions, senescence is associated to G1-G1/S phase arrest. Transition from G1 to S phase is controlled by the activity of the transcription factor E2F. In normal scenarios, Rb protein forms a complex with E2F, inhibiting its activity and therefore preventing cell cycle to continue. When cyclin dependent kinases (CDKs) phosphorylate Rb protein, E2F is released. However, in the senescence state, CDKs are inhibited by different upstream cues, such as p16 or p21, which also prevents Rb phosphorylation and subsequent cell cycle progression.

This has been corrected in the revised version of the manuscript.

10. Line 247: The link between “elimination of senescent cells improves these pathologies [such as liver fibrosis]” and “Krizhanovsky showed that hepatic stellate cells” “undergo senescence” “enhancing the expression of the matrix metalloproteases with fibrolytic activity” “hence limiting liver fibrosis” seems to argue for the opposite”

Reviewer #1 is totally right with their comment. Hepatocytes and cholangiocytes that have undergone senescence promote liver fibrosis, mainly by inducing hepatic stellate cell activation and in turn, extracellular matrix deposition. However, as Krizhanovsky *et al.* demonstrated, hepatic stellate cells can have an opposite effect and induce fibrosis regression (Krizhanovsky *et al* Cell 2018). It has been corrected and discussed in detail in the manuscript in **lines 327-337**.

11. Line 428: As above, p38MAPK is not an exclusive feature of dormant cells, but a central signaling cascade active in senescent cells (see Freund-A *et al.*, EMBO J, 2011) “ in other words, the distinction of dormant, quiescent and senescent cells remains blurry, potentially for the reason that these terms may actually describe largely similar conditions. Table 1 is not sufficient to elucidate the problem “ it”s rather the missed aspects (e.g. genomic re-organization, alterations of the nuclear envelope, the expanded lysosomal compartment a.o.), which might help to distinguish and to conclude structure-to-function implications if those cells re-enter the cell-cycle.

As correctly pointed out by Reviewer #1, p38 MAPK and/or ERK activity are not exclusive for quiescent cancer cells (Freund *et al*, EMBO J 2011). However, although mentioned in the section “Strategies to Target Dormant Quiescent Cancer Cells”, we have stated that the strategy of modulating p38/ERK was a general feature of dormant cancer cells (both senescent and quiescent cancer cells).

Nevertheless, the abovementioned section has been re-structured and rewritten in order to clarify our claims. A brief definition of dormancy has been added (line 132). We have also added the following important features both in the Table 1 and in the manuscript in **lines 238-245 and 269-278**.

- ***SAHF formation and γ H2AX foci.*** In fact, SAHF plays a role in sequestering and silencing genes needed for proliferation.
- ***Lysosomal compartment expansion.*** It leads to an increase in SA- β -gal activity.
- ***Cytokinetic block*** caused by p16INK4a-Rb and mitogenic pathways, which supports irreversible cellular arrest of senescent cells.

REVIEWER #2

In this manuscript Santos-de-Frutos et al review the current status of research in the area of tumor dormancy. This is a very thoroughly researched, well-written and timely review focusing on the role of therapy, senescence and therapy induced senescence in tumor dormancy. While both therapy and senescence have been postulated to have a potential role in the induction/maintenance/exit from dormancy, proof-of concept studies in these areas are lacking. This review provides a comprehensive overview of the current status of ongoing research in this field and will influence thinking in this field. However, there are a few minor concerns.

There are few grammatical errors and typos that needs fixing.

The abstract needs to focus more on the role of senescence and therapy in the dormancy program rather than a generalized abstract about dormancy.

Although the review focuses on the role of senescence in dormancy program, the additional sections discussing the anti- and pro-tumorigenic roles of senescence distracts away from the emphasis on its role in dormancy. While there are plenty of reviews examining the role of senescence in the former, not many discuss the role of this important physiological program in regulating tumor dormancy. The authors therefore need to reconsider the addition of these sections to the manuscript.

We are very grateful to Reviewer #2 for their general interest on our manuscript and for finding it a thoroughly researched review and appropriate for Communications Biology. We also thank this Reviewer for the comments to improve our manuscript. Accordingly, we addressed the following concerns:

- Grammatical errors and typo found in the manuscript have been corrected
- The abstract has been modified according to suggestions from Reviewer #2
- Regarding the addition of a section of the importance of senescence in regulating dormancy, we believe that we have mentioned it in the section “Dormant Senescent Cancer Cells”, specifically starting at **line 279**. We believe that senescent cells are important in regulating tumour dormancy mainly by SASP secretion, which leads to infiltration of different immune cells including NK cells, neutrophils, monocytes/macrophages and T cells. SASP content will definitely determine how immune cells respond, either promoting the clearance of tumour cells or protecting them from immunosurveillance. Therefore, senescent cells will induce and regulate dormancy when controlling immune cells from eliminating dormant cancer cells.

REFERENCES

- Agudo, J. et al. (2018) Quiescent Tissue Stem Cells Evade Immune Surveillance. *Immunity* 20 (48), 271-285.e5.
- Aguirre-Ghiso, J.A. (2007) Models, mechanisms and clinical evidence for cancer dormancy. *Nat Rev Cancer* 7 (11), 834-846.

Clarke, M.F. et al. (2006) Cancer Stem Cells - Perspectives on Current Status and Future Directions: AACR Workshop on Cancer Stem Cells. *Cancer Res* 66 (19), 9339-9344.

Freund, A. et al. (2011) p38MAPK is a novel DNA damage response-independent regulator of the senescence-associated secretory phenotype. *EMBO J* 30 (8), 1536-1548.

Iannello, A. et al. (2013) p53-dependent chemokine production by senescent tumor cells supports NKG2D-dependent tumor elimination by natural killer cells. *J Exp Med* 210 (10), 2057-2069.

Krizhanovsky, V. et al. (2008) Senescence of Activated Stellate Cells Limits Liver Fibrosis. *Cell* 134 (4), 657-667

Krug, U. et al. (2016) Increasing intensity of therapies assigned at diagnosis does not improve survival of adults with acute myeloid leukemia. *Leukemia* 30 (6), 1230-1236.

Malladi, S. et al. (2016) Metastatic Latency and Immune Evasion through Autocrine Inhibition of WNT. *Cell* 165 (1).

Milanovic, M. et al. (2018) Senescence-associated reprogramming promotes cancer stemness. *Nature* 553 (7686), 96-100.

Mosteiro, L. et al. (2016) Tissue damage and senescence provide critical signals for cellular reprogramming in vivo. *Science* 354 (6315), aaf4445.

Saito, Y. et al. (2010) Induction of cell cycle entry eliminates human leukemia stem cells in a mouse model of AML. *Nat Biotechnol* 28 (3), 275-280.

REVIEWERS' COMMENTS:

Reviewer #1 (Remarks to the Author):

This is now the revised version of the manuscript "When dormancy fuels tumour relapse" by Santos-de-Frutos and Djoudner.

In their point-by-point rebuttal, the authors attempted to address all the concerns raised in my first statement to their initial submission. I acknowledge that the manuscript is now more balanced and contains additional important references.

However, the authors remain vague in their response to many points, and often, the explicitly named sections ("lines xxx-xxx") in the revised version do not contain a reflection of the novel thoughts discussed in the rebuttal, but rather a loose connection of conceptual ideas. For instance, the issue of feasible cancer dormancy maintenance therapies is not even remotely addressed in lines 524-528 and lines 593-599.

Also, I sometimes simply do not understand the way of thinking. As an example, I brought up the point that in many tumor entities relapses occur rather early than late (with the latter reflecting the idea of long-term dormant cancer cells giving rise to a relapse). I don't know what to make out of the authors' statement that "there is no strict definition regarding the time for disease recurrence". Along those lines, consultation of a clinical oncologist may have helped, but the authors argued against.

Detailed responses to the Reviewers' comments

REVIEWER #1

This is now the revised version of the manuscript "When dormancy fuels tumour relapse" by Santos-de-Frutos and Djouder. In their point-by-point rebuttal, the authors attempted to address all the concerns raised in my first statement to their initial submission. I acknowledge that the manuscript is now more balanced and contains additional important references.

We are very grateful to Reviewer #1 for considering that the modifications made have improved the manuscript. We are also thankful to this Reviewer for pointing out their concerns in order to improve and clarify all the points mentioned in the review.

As requested, we have now introduced the latest textual changes in the manuscript and highlighted them in green.

We hope that the new changes meet Reviewer #1's expectation.

However, the authors remain vague in their response to many points, and often, the explicitly named sections ("lines xxx-xxx") in the revised version do not contain a reflection of the novel thoughts discussed in the rebuttal, but rather a loose connection of conceptual ideas.

We are thankful to this Reviewer for pointing out this.

Based on their concerns, we have modified the manuscript in order to better reflect our thoughts as discussed in the previous point-by-point responses.

For instance, the issue of feasible cancer dormancy maintenance therapies is not even remotely addressed in lines 524-528 and lines 593-599.

This part has been better discussed in lines 556-565 and lines 627-628.

Also, I sometimes simply do not understand the way of thinking. As an example, I brought up the point that in many tumor entities relapses occur rather early than late (with the latter reflecting the idea of long-term dormant cancer cells giving rise to a relapse). I don't know what to make out of the authors' statement that "there is no strict definition regarding the time for disease recurrence". Along those lines, consultation of a clinical oncologist may have helped, but the authors argued against.

We apologize if Reviewer #1 has found the modifications made in the abovementioned part confusing or unclear.

As requested, we have better discussed the concepts of early and late relapses in the manuscript in lines 111-122.